# Models and Algorithms for the Refinement of Therapeutic Approaches for Retinal Diseases

**DOI:** 10.3390/diagnostics13050975

**Published:** 2023-03-03

**Authors:** Elfriede Friedmann, Simon Dörsam, Gerd U. Auffarth

**Affiliations:** 1Institute of Mathematics, Kassel University, Heinrich-Plett-Straße 40, 34132 Kassel, Germany; 2Department of Ophthalmology, David J. Apple International Laboratory for Ocular Pathology, Heidelberg University, 69120 Heidelberg, Germany

**Keywords:** numerical modeling, AMD treatment, drug distribution, drug diffusion, free convection, partial differential equations, finite element methods

## Abstract

We are developing a Virtual Eye for in silico therapies to accelerate research and drug development. In this paper, we present a model for drug distribution in the vitreous body that enables personalized therapy in ophthalmology. The standard treatment for age-related macular degeneration is anti-vascular endothelial growth factor (VEGF) drugs administered by repeated injections. The treatment is risky, unpopular with patients, and some of them are unresponsive with no alternative treatment. Much attention is paid to the efficacy of these drugs, and many efforts are being made to improve them. We are designing a mathematical model and performing long-term three-dimensional Finite Element simulations for drug distribution in the human eye to gain new insights in the underlying processes using computational experiments. The underlying model consists of a time-dependent convection-diffusion equation for the drug coupled with a steady-state Darcy equation describing the flow of aqueous humor through the vitreous medium. The influence of collagen fibers in the vitreous on drug distribution is included by anisotropic diffusion and the gravity via an additional transport term. The resulting coupled model was solved in a decoupled way: first the Darcy equation with mixed finite elements, then the convection-diffusion equation with trilinear Lagrange elements. Krylov subspace methods are used to solve the resulting algebraic system. To cope with the large time steps resulting from the simulations over 30 days (operation time of 1 anti-VEGF injection), we apply the strong A-stable fractional step theta scheme. Using this strategy, we compute a good approximation to the solution that converges quadratically in both time and space. The developed simulations were used for the therapy optimization, for which specific output functionals are evaluated. We show that the effect of gravity on drug distribution is negligible, that the optimal pair of injection angles is $\left(50{}^{\circ },50{}^{\circ }\right)$, that larger angles can result in 38% less drug at the macula, and that in the best case only 40% of the drug reaches the macula while the rest escapes, e.g., through the retina, that by using heavier drug molecules, more of the drug concentration reaches the macula in an average of 30 days. As a refined therapy, we have found that for longer-acting drugs, the injection should be made in the center of the vitreous, and for more intensive initial treatment, the drug should be injected even closer to the macula. In this way, we can perform accurate and efficient treatment testing, calculate the optimal injection position, perform drug comparison, and quantify the effectiveness of the therapy using the developed functionals. We describe the first steps towards virtual exploration and improvement of therapy for retinal diseases such as age-related macular degeneration.

## 1. Introduction

Retinal diseases are unfortunately the most common cause of blindness in wealthy countries and already the most common cause of childhood blindness worldwide [1,2,3,4,5,6,7,8,9,10]. In people with retinal diseases, the light-sensitive neural cells in the retina are damaged and, in the worst case, die. For a long time, little attention was paid to these diseases, partly because they came to the fore late, and partly because in the past there were no treatment options. This has changed drastically today and although there are still forms of retinal degeneration for which there is no cure, it has been shown for many age-related eye diseases that prevention is possible and successful treatment can also be available. Accurate diagnostics and the resulting treatment require modern and expensive equipment. Extraordinary progress has been made in research by integrating information from animal and tissue culture models with clinical observations and with retinal biochemistry and physiology [11,12,13,14,15,16,17,18]. Artificial Intelligence research has shown great promise, especially in the classification of diagnostic images in this medical area [19,20].

Age-related macular degeneration (AMD) is the retinal disease that is the most common cause of vision loss in industrialized countries. It is a disease of the macula that results from late-onset neurodegeneration of the pigment epithelium photoreceptor complex [21]. The disease affects 10% of those over 65 years of age and 25% of those over 75 years of age. The characteristic sign of AMD is drusen in the macula. The exact pathology is not yet fully understood, but it is thought to be a complex interaction of many factors. The formation of drusen promotes loss of the retinal pigment epithelium (RPE), dysfunction of Bruch’s membrane and further photoreceptor death. Progressive damage of Bruch’s membrane along with upregulation of vascular endothelial growth factor (VEGF), a biomarker for AMD, leads to uncontrolled growth of abnormal choroidal vessels under the RPE [21]. There is no standard treatment for dry AMD, and several innovative treatments are in progress. For wet AMD significant advances has been made in recent years. Prior to the introduction of the first anti-VEGF agent (Macugen) around 2004, laser photocoagulation and photodynamic therapy were used. Since then, other more effective anti-VEGF agents have been developed: Lucentis (ranibizumab), Avastin (bevacizumab), Eylea (aflibercept), and the latest approved agents Beovu (brolucizumab, 2020) [22] and Vabysmo (faricimab, 2022) [23]. Treatment is not fixed; there are four dosing regimens in clinical use [21]. Monthly or bimonthly injections lead to long-term success, but do not take into account individual disease progression. A pro-re-nata strategy requires many check-ups and does not show good improvement, as reactivation of the disease is often detected too late, which can lead to irreversible visual loss. A treat-and-extend regimen is a personalized treatment in which the next injection is optimally planned [24,25]. An observe-and-plan regimen is also a personalized treatment that includes a personalized treatment plan with multiple injections [26]. Of course, the newly developed drugs are more effective and therefore require fewer injections. There are still many unresolved issues regarding frequency of treatment, proper dose, location of injection, effectiveness of therapy, choice of drug and more. Nevertheless, due to the high cost of the drugs and the inconvenience of the injections, as well as the increasing burden on patients due to the many consultations and inconvenient injections, there is an urgent need for a personalized, long-lasting optimized therapy solution for patients with AMD. One promising approach is treatment with a Port Delivery System filled with a drug that can be refilled [27] or with drug loaded hydrogels for sustained delivery [28,29].

In this article, we use mathematical modeling and numerical simulation to refine the results of the therapeutic approach for AMD. These so-called in silico or *virtual* experiments cost less than laboratory experiments, are not limited in performance, and do not have to follow ethical rules. Once developed and implemented, they can be used as an additional source of information in combination with experiments for parameter identification or validation. They can help accelerate and improve insights for therapeutic approaches. In addition, we have developed a model and algorithms that can also be used for various retinal or other diseases or other mass transfer problems in the eye.

Initial studies on the treatment of age-related macular degeneration based on computer simulations can be found in [30,31,32,33]. In [30], the influence of the diffusion coefficient, retinal permeability, and vitreous humor flow on drug distribution was analyzed. The results showed that for rapidly diffusing drugs (drugs with high diffusion coefficients), aqueous humor convection plays a minor role in drug transport. For slow diffusing drugs (drugs with small diffusion coefficients) and low viscosity vitreous fluid, convection plays a greater role and may result in higher drug concentrations reaching the retina. These higher drug concentrations in one location can generally be potentially toxic. Each drug must be evaluated in this regard. Delayed injection of the drug has been shown to avoid conditions of retinal toxicity and allow lower drug concentrations with a longer residence time on the retinal surface. Drugs with high diffusivity and retinal permeability cause uniform distribution of the drug along the retinal surface, whereas drugs with low diffusivity and retinal permeability localize the drug concentration along the posterior retinal surface. The mathematical model uses the Navier Stokes equations for aqueous humor flow coupled to the convection diffusion equation used for drug distribution. Here, an additional term was used for release of the drug from the injection site at a specific rate. The simulations were performed using Ansys Fluent [34]. In [32], the concentration distribution of the drug around the macula of a rabbit eye was analyzed. For this purpose, the geometrical model and some of the parameters of the mathematical model were calibrated with respect to experimental results with rabbit eyes. Running a model and simulations together with experiments has the advantage that the model is calibrated against the experiments and in a sense validated. Two substances were considered in the experiment: Fluorescein and FITC-Dextran. Fluorescein has a larger diffusion coefficient, $D_{F}=6\times 10^{-10}\;$
$m^{2}$/s, than FITC-Dextran,
$D_{D}=3.9\times 10^{-11}\;$
$m^{2}$/s, and therefore diffuses much faster, and higher fluorescein concentrations are found around the macula. In the experiments and simulations the drug distribution was observed for one day. Three simulation results are shown: the drug distribution after 5, 15 and 24 h. The results were that fluorescein reaches the macula earlier and in higher concentration, towards the end of the day nothing arrives at the macula because fluorescein is depleted. The behavior of FITC-Dextran concentration is different, due to the slower diffusion, the concentration around the macula builds up slowly but steadily. The diffusion coefficient of FITC-Dextran has a similar magnitude to the diffusion coefficient of bevacizumab [35] measured in a rabbit eye. Simulations were performed using a finite volume method. In [33], the effects of injection time, needle gauge, and needle angle are analyzed using an older, simpler mathematical model. Here, simulations in three dimensions (3D) are presented for the first time in a simple idealized geometry. Previously, only two-dimensional simulations had been performed. The distribution of fluorescein in the vitreous is also analyzed. Two injection angles are compared in the plane of the optical axes. The result was that more drug reaches the macula when the injection is in the direction towards the macula. Depending on the injection, both the toxicity risk and the drug effect may increase or decrease. The 3D geometry is roughly estimated and complex to construct. We will present a solution for personalized geometry construction that can be easily adapted to patient data (OCT and US).

In this paper, we present an extended model for drug distribution in the human eye. Our mathematical model is based on the approach in [36]. There, a transport-diffusion equation was used to describe drug distribution in the vitreous, and it was coupled with the Darcy equation for vitreous humor flow. We use the same equations in 3D (large computational cost and higher effort for grid construction) with a different inflow condition for a better physical representation of the flow, we add gravity to discuss its effect on drug distribution, we add anisotropic diffusion by including collagen fibres and we add nonlinear diffusion for certain substances. First, we present a realistic geometry of the vitreous body. For the geometry description in three dimensions, we use mathematical functions fitted to our own experimental data. This is an advantage for the numerical simulations, the grid can be constructed for the desired fineness. All previous models in the cited papers have not taken gravity into account, with the exception of [37]. Here, experiments have shown that gravity should play a role in drug distribution. An unusually strong inflow profile was also used there. We systematically analyze the influence of gravity on drug distribution in the vitreous and complete the model with the correct inflow conditions in the ciliary body region. First results of our simulations in two dimensions are presented in [31]. In the following, we discuss the differences in the simulation methods and in the obtained results compared to previous models. In addition, the success of the therapy is investigated with respect to various optimization parameters.

## 2. Materials and Methods

### 2.1. Mathematical Models

We have developed a method to construct a patient-specific vitreous body from an ultrasound scan. To describe the shape of the vitreous body, we have developed a mathematical formula that describes a modified Limacon. The formula contains parameters that are adjusted to the ultrasound data (see Figure 1). For ultrasound data, B-scans were generated from left to right and from right to left. From every image 164 data points were extracted. The data set tables can be found in Appendix A, Table A1,Table A2,Table A3,Table A4. All data points are used to perform the parameter estimation to determine the shape of the vitreous body. Three of the parameters of the mathematical model can be adjusted to personalize the vitreous shape. As proof of concept we used all quantitative data sets from 12 patients, each with 164 data points, to describe an average vitreous shape that we use for our “Virtual Eye” in the computer. The maximum deviation of the data is 0.34 mm. More ultrasound data from a large number of patients may be included in a future study. The methods developed can be easily applied to more data. Due to rotational symmetry, it is sufficient to describe the two-dimensional profile of the vitreous:(1)$$x=R\left(q,\phi \right)\cos\left(\phi \right)+m_{x},\;y=R\left(q,\phi \right)\sin\left(\phi \right),\;R\left(q,\phi \right)=q_{1}+q_{2}\cos\left(\hat{\phi }\right)+q_{3}\cos{\left(\hat{\phi }\right)}^{3},$$
where $q=\left(q_{1},q_{2},q_{3}\right)\in R^{3}$, $\left(m_{x},0\right)$ is the center of the Limacon, $m_{x}\in R$ and ϕ∈[0,2π]. With $\hat{\phi }$ we denote the dependency on the parameter $m_{x}$ after converting the data into polar coordinates: $\phi \left(m_{x}\right)=\arccos\left(\frac{x-m_{x}}{r}\right)$ with $r=\sqrt{{\left(x-m_{x}\right)}^{2}+y^{2}}$. For more information, see [38,39].

Next, we describe the physiology in the vitreous body of the human eye. We begin with the flow of the aqueous humor produced in the ciliary body. Most of this fluid circulates in the anterior chamber of the eye and is drained through the trabecular meshwork into Schlemm’s canal. A smaller portion flows through the vitreous body, enters the retina, and is flushed out through the blood system. We describe the flow of aqueous humor with the Darcy model [36,40]:(2)$$v=-\frac{\kappa }{\mu }\nabla p,\;\mathrm{div}\;v=0,$$
where κ is the hydraulic conductivity, μ the viscosity of the fluid, *v* its velocity and *p* the pressure. The Darcy model describes flow in porous media and represents a first approximation to the physiology in the human vitreous, since the flow penetrates a viscoelastic medium containing a collagen network. In terms of boundary conditions, we know that the vitreous is bounded by the retina and the lens. Between the lens and the retina, the aqueous humor has room to enter through the hyaloid membrane. We denote the retinal boundary by $\Gamma_{R}$, the lens boundary by $\Gamma_{L}$ and the inflow boundary by $\Gamma_{H}$. Thus, the boundary of Ω is $\partial \Omega =\Gamma_{R}\cup \Gamma_{L}\cup \Gamma_{H}$. We assume that the lens is an impermeable organ so that aqueous humor cannot penetrate. Thus, we obtain homogeneous Neumann boundary conditions at the lens:(3)$$n\cdot v=0\;\;\mathrm{on}\;\Gamma_{L},$$
where *n* is the normal vector. In the literature, the boundary conditions for the Darcy equations are generally Dirichlet conditions for pressure. Our simulations have shown that the flow near the inflow looks unphysical due to our mixed boundary conditions, so we impose a Poiseuille velocity profile for the inflow:(4)$$n\cdot v\left(x,y,z\right)=c_{pflow}\left(R_{pflow}^{2}-r_{pflow}^{2}\left(x,y,z\right)\right)=:v_{pflow}\;\;\mathrm{on}\;\Gamma_{H},$$
where $c_{pflow}$ is a parameter that regulates the strength of the inflow, $R_{pflow}$ is half the distance from the lens to the retina, and $r_{pflow}$ is the distance from a velocity particle to the central circular ring between the lens and retina.

The epiretinal membrane is the outer layer of the retina connected to the vitreous body. The flow of a fluid through this membrane is described with a permeability condition and is a Robin-type boundary condition:(5)$$n\cdot v=K_{RCS}\frac{p-P_{v}}{L}\;\;\mathrm{on}\;\Gamma_{R},$$
where $P_{v}$ is the episcleral pressure, $K_{\mathrm{RCS}}$ the total hydraulic conductivity and *L* the thickness of the retina.

Now we present the model of the anti-VEGF treatment of age related macular degeneration. We skip the injection process and start with long-term simulations for the drug distribution when the drug is already in the vitreous. The drug distribution is given by the following transport-diffusion equation:(6)$$\partial_{t}C+\left(v\cdot \nabla \right)C-D\Delta C=0$$
where *C* is the drug concentration, which here depends on space and time t∈[0,T] and *D* is the diffusion parameter of the drug in the vitreous. We use $D=4\times 10^{-11}\;$
$m^{2}$/s for bevacizumab in a rabbit eye from [35] in our model for the human eye model. In our simulations, we consider only one injection, i.e., we simulate the drug distribution during one month. At the lens we assume homogeneous boundary conditions, the drug does not penetrate the lens:(7)$$\partial_{n}C=0\;\;\mathrm{on}\;\Gamma_{L}\times \left[0,T\right].$$

At the retina we have a Robin type boundary condition
(8)$$PC+\left(n\cdot v\right)kC=-D\partial_{n}C+\left(n\cdot v\right)C\;\;\mathrm{on}\;\Gamma_{R}\times \left[0,T\right],$$
where *k* is a partition coefficient for the drug describing the fraction in the vitreous and retina, and *P* is the permeability of the retina. At the hyaloid membrane, we assume a homogeneous Dirichlet boundary condition C=0 in this model, which means that the drug will not diffuse against the flow. In another project involving pharmacology in the model [41], the concentrations of the different complexes may leak into the anterior part of the eye through the boundary at the hyaloid membrane between the anterior chamber and the vitreous.

### 2.2. Numerical Methods

The presented model (Equation 2)–(Equation 8) is solved numerically by the Finite Element (FE) method using the C++ software library deal.ii [42,43]. In our model, the Darcy equation is time independent and can be decoupled from the system. It is first solved using the mixed Finite Element method, then the computed velocity can be substituted into the time-dependent convection-diffusion equation, which is discretized using the Rothe method [44]. In the spatial discretization, we use trilinear Lagrange elements for the drug concentration, Raviart-Thomas elements for the velocity, and discontinuous piecewise constant elements for the pressure. To solve the time dependent model with sufficient accuracy, the time-stepping method should be at least second order and stable, so the fractional step theta scheme is used. A detailed description can be found in [45].

The resulting system of algebraic equations is transformed into a saddle point problem in the Darcy case:(9)$$\left(\begin{array}{cc} M & B\\ B^{T} & 0 \end{array}\right)\left(\begin{array}{c} V\\ P \end{array}\right)=\left(\begin{array}{c} F\\ 0 \end{array}\right),$$
where *M* is the mass matrix, *B* is the matrix resulting from the discretization of the divergence, *V* is the vector with the degrees of freedom of the velocity, *P* is the vector with the degrees of freedom of the pressure, and *F* is the vector of the right hand side. A detailed derivation of this linear system of equations is described in [39]. The reason of this transformation into a saddle point problem is that often the initial problem is ill-conditioned or very poorly conditioned, and cannot be solved directly. Therefore, the problem is represented as a saddle point problem. This form of representation does not change the properties of the system like invertibility, spectral properties, and conditioning, these are maintained, but there is a well developed solution theory for this representation. Exact knowledge of the system properties is important for the development of solution algorithms. In some cases, the special structure of the saddle point problem can be exploited to avoid or mitigate the ill-conditioning. The structure of the right-hand side also plays a role here. A common solution method for (Equation 9) is the Schur complement technique, a segregated approach, which is also very well suited for our problem. The difficulty here is that there is no so-called best method for all kinds of problems, but different efficient solvers, which were developed for certain model equations. Some solution methods are presented in [46]. With the Schur Complement method one obtains the following system of linear equations:(10)$$\begin{array}{cc} M_{h}V_{h} & =F_{h}-B_{h}P_{h},\\ B_{h}^{T}M_{h}^{-1}B_{h}P_{h} & =B_{h}^{T}M_{h}^{-1}F_{h}. \end{array}$$

The matrix $B^{T}M^{-1}B$ is here the Schur complement, and is symmetric and positive definite. The Conjugate Gradient (CG) method [47] can be used to solve the system of linear Equations (Equation 10).

The fully discretized convection-diffusion equations for each time step are as follows:(11)$$\left(M_{h}^{C}+\theta \Delta tA_{h}\right)C_{h}^{m}\left(t\right)=\left(M_{h}-\left(1-\theta \right)\Delta tA_{h}\right)C_{h}^{m-1}\left(t\right),$$
where $M_{h}^{C}$ is the mass matrix and $A_{h}$ the stiffness matrix, *C* is the vector with the degrees of freedom for the concentration, θ and Δt are parameters given by the fractional step theta scheme. In this case, the resulting linear system has a nonsymmetric matrix. Iterative methods such as GMRES with ILU preconditioning work well with a large sparse matrix [48], i.e., when a solution is computed on a fine grid, and are also generally not affected by a single zero eigenvalue. Without preconditioning, the iterative Krylov subspace method converges poorly. Preconditioning is a simple transformation of the linear system and leads to a coefficient matrix with the required spectral property that all eigenvalues are contained in the half-plane Re(z)>0, z∈C, i.e., it is non-singular.

For the numerical implementation of our model, more than 6000 lines of code were implemented in the existing deal.ii environment. The most time-consuming part of the solution process is setting up and solving the algebraic system of equations. The code was validated and a convergence analysis was performed.

### 2.3. Mathematical Functionals for Drug Comparison

With the methods developed, we are studying the effect of injection position in terms of how much drug remains in the vitreous and how much drug operates in a specific region. To quantify these effects, output functionals are developed and included in [38].

The functional $J_{\Omega }\left(t,C\right):R^{+}\times H_{0}^{1}\left(\Omega \right)\to R^{+}$ denotes the relative amount of drug *C* remaining in the vitreous at the current time,
(12)$$J_{\Omega }\left(t,C\right):=\frac{\int_{\Omega }C\left(t,x\right)\;dx}{\int_{\Omega }C\left(0,x\right)\;dx}.$$

The functional $J_{M}\left(t,C\right):R^{+}\times H_{0}^{1}\left(\Omega \right)\to R^{+}$ denotes the amount of drug *C* present in a specific region *M* at the current time,
(13)$$J_{M}\left(t,C\right):=\int_{M}C\left(t,x\right)\;dx,\;M=B_{r}\left(m\right)\cap \Omega ,$$
where $B_{r}\left(m\right)$ is a sphere with center *m* at the macula and radius *r*. The drug is one of the antibody therapies and blocks VEGF by binding to it and washing it out, preventing the formation and growth of vessels around the macula. In our model, we focus on the drug distribution by diffusion and convection. The kinetics of the drug is the subject of a separate project and will be presented in a future paper. Here, we consider only the drug transport to the macula and estimate the amount of drug that reaches this area without reactions in the vitreous. Depending on the stage of the disease, VEGF may already be distributed in the vitreous. In this case, the drug is already acting there.

Finally, we estimate the total amount of drug that can react with VEGF molecules in a period [0,T]: $J_{M,\Omega }\left(T,C\right):R^{+}\times H_{0}^{1}\left(\Omega \right)\to R^{+}$
(14)$$J_{M,\Omega }\left(T,C\right):=\int_{0}^{T}\frac{\int_{M}C\left(t,x\right)\;dx}{\int_{\Omega }C\left(0,x\right)\;dx}\;dt,\;M=B_{r}\left(m\right)\cap \Omega ,$$
where $B_{r}\left(m\right)$ is a sphere with the center *m* at the macula and radius *r*.

Due to the integration over time, the amount of drug reaching the macula is overestimated because it takes time for the drug to diffuse into this area as well. In addition, the amount depends on the choice of the size of $B_{r}\left(m\right)$. In this paper, we choose r=2 mm. Furthermore, we can measure how much of the drug does not reach this area and is lost by diffusion through the retina during the period [0,T]: $J_{R}\left(T,C\right):R^{+}\times H_{0}^{1}\left(\Omega \right)\to R^{+}$
(15)$$J_{R}\left(T,C\right):=1-J_{\Omega }\left(T,C\right)+J_{M,\Omega }\left(T,C\right).$$

These functionals are used for the drug comparison study.

## 3. Results

We presented long-term finite element simulations of drug distribution in the vitreous that include a period of 30 days. This period corresponds to the typical time from the first to the second injection with, for example, ranibizumab. In Figure 2 we visualize the 3D drug distribution in our Virtual Eye and the aqueous humor flow. The flow is produced at the ciliary body, enters in the vitreous via the zonules and leaves the vitreous through the retina via permeability. The drug is distributed through diffusion and convection. At the initial time there is a ball shaped concentration distribution near the lens (the injected drug) and at the presented time already some diffusion and convection occured which is shown in the Figure.

### 3.1. The Influence of Gravity on Drug Distribution

It is assumed that the patient’s head position after injection has a relevant influence on drug distribution in the eye. The experiments in [37] confirmed the effect of gravity on the distribution of bevacizumab in an undisturbed balanced salt solution in vitro. Bevacizumab did not immediately dissolve and distribute evenly in the solution as expected, but rather settled in the lower part of the tube than in the upper part due to gravity. This effect was still observed after 7 days. Thus, whether the patient is standing or lying down may matter when a rapid local effect is needed.

In our simulations we consider a patient with age-related macular degeneration in the left eye. The drug is injected from the left side. For simplicity, we assume that the patient has the usual head orientation, i.e., the face is directed forward. The drug distribution is computed over 30 days for the following cases:The patient stands (over the total time).The patient lies on the back.The patient lies sideways on the left side.The patient lies sideways on the right side.The patient stands half the day and lies on the back for the rest of the day.

The results of our simulations are shown in Figure 3. The influence of gravity turns out to be small. When the patient is lying on the right side, 38.4% of the injected drug reaches the macula, and we calculate the highest concentration of $0.287\frac{\mathrm{kg}}{m^{3}}$ around the macula at 6 days after injection. In all other cases, slightly less drug reaches the macula. The worst case, if any, is when the patient is lying on the left side where the drug was injected. Then 37.6% of the injected drug reaches the macula. The highest concentration is $0.281\frac{\mathrm{kg}}{m^{3}}$. To deliver more drug to the macula, it is advantageous if gravity points in the direction of injection. In the other three cases, we observe a positive effect when gravity is directed toward the macula. We measure 38.3% of the injected drug on the macula when the patient is lying on its back, 38% when the patient is standing, and 38.2% when the patient stands half the day and lies on its back the rest of the day.

In summary, patients lying on their back or on the injection side showed the most successful therapy, but in our models and simulations gravity does not play a significant role. The results show a maximum difference of less than 1% in the concentration of injected drug at the macula. However, we have considered a healthy homogeneous vitreous here. In a heterogeneous vitreous with a more complex consistency, the situation may change.

### 3.2. Optimal Injection Position

The goal of an optimal treatment of age-related macular degeneration is the local effect of the drug as a VEGF blocker at the macula. The drug should act in this way for as long as possible to achieve the best results. The therapy is expensive and the injection is uncomfortable, so the drug should be used effectively. Furthermore, too high drug concentrations can lead to toxicity. Thus, only certain doses of the drug are injected over several weeks.

In this section, we will analyze whether the healing process, as measured by the amount of drug reaching the macula, depends on the injection position. Regardless of a feasible initial concentration we will investigate which location is optimal for the injection. Four different injection positions (see Figure 4) are considered as toy problems (virtual experiments) to find out the relevance of the injection position. We do not discuss whether the chosen positions are realistically reasonable. The first position we consider is the standard position for an injection into the left eye: 3.5 mm from limbus across the pars plana toward the center of the posterior pole (in geometry: 3.5 mm from the limbus and 5.5 mm to the left of the pupil axis). The needle depth is 5 mm. The second position we consider is 10 mm from the limbus via the pars plana direction to the center of the posterior pole. Here the needle penetrates to the center of the vitreous (∼10 mm). The third injection position is 8 mm from the limbus via the pars plana direction laterally to the posterior pole. The optic nerve serves as a point of orientation. The needle penetrates ∼7–8 mm. The final injection position we will analyze is chosen 10 mm from the limbus via the pars plana direction toward the center of the posterior pole. Here, the needle penetrates in the direction of the optic nerve (∼18 mm).

A period of 30 days is used for all four simulations. The results are shown in Figure 5. The standard injection always results in the lowest concentration on the macula. This indicates that the therapy can be optimized. To achieve the highest drug concentration around the macula, the most obvious option is the injection directly near the macula. This goal is achieved only in the first eight of 30 days. After nine days, injection into the central vitreous leads to the highest drug concentration at the macula. At the second injection position, the drug remains around the macula the longest. At the fourth injection position, the highest amount of drug (10 times higher) reaches the macula.

The pathology and permeability of the vitreous as well as of the retina, and the injection site have a major impact on drug distribution. Injection in the second position is the best choice for longer acting drugs, e.g., aflibercept: these drugs work best when injected into the center of the vitreous, they must be effective for several weeks. Injection in the fourth position is the best choice for intensified initial treatment, such as the treatment of a thick edema: In the first 50 h, we have the highest amount of drug in the macula area (more than 10× higher), but after 50 h, the drug concentration is lower than with the injection at the second position, which is close to the vitreous border and the retina, where part of the drug escapes. The standard position also has the disadvantage of allowing some of the drug to escape through the zonule into the anterior chamber, resulting in a relatively smaller amount of drug reaching the macula. However, exactly how much reaches the macula depends on the completeness of the model and the accuracy of the parameters used.

### 3.3. Optimal Injection Angle

This subsection analyzes the influence of the injection angle on the amount of drug reaching the macula, the site of action. Although there are precise instructions for the injection procedure, position and needle length, in practice there are slight differences in position and penetration angle for each individual injection. In this section, we will perform some virtual (toy) experiments to gain insights into the sensitivity of the amount of drug at the macula with respect to the angle of injection. We will discuss injection positions that may not be appropriate from a medical perspective, but help provide detailed information about the physics and physiology behind the injection process and whole treatment.

We introduce a coordinate system to define two penetration angles $\psi_{xy}$ and $\psi_{z}$ for orientation in space, see Figure 6. In our geometry, the x-axis is the optical axis and points from the lens to the retina (in Figure 6, it points upward), the y-axis is the horizontal line perpendicular to the optical axis, running from left to right in Figure 6, and the z-axis is the vertical line perpendicular to the optical axis, pointing in Figure 6. We define the penetration angle $\psi_{xy}$ between the needle direction and the optical (x-) axis on the xy-plane and the penetration angle $\psi_{z}$ between the needle direction and the z-axis.

In [33], the standard injection was defined as $\psi_{xy}=50{}^{\circ }$ and compared to the penetration angle ${\tilde{\psi }}_{xy}=75{}^{\circ }$. The penetration angle $\psi_{z}$ was set to $90{}^{\circ }$ in both cases. In our simulations, we analyze $\psi_{xy}=50{}^{\circ }$, ${\tilde{\psi }}_{xy}=75{}^{\circ }$ and additionally ${\hat{\psi }}_{xy}=25{}^{\circ }$ and we include several other possibilities with $\psi_{z}\neq 90{}^{\circ }$, i.e., the needle is not injected along the xy-plane. For comparison we calculate the functionals $J_{M}$, $J_{R}$, $J_{M,\Omega }$, $J_{\Omega }$ from Section 2.3 for each simulation configuration. First, we analyze the injections with $\psi_{z}=90{}^{\circ }$ following [33]. Our simulations confirm the results in [33], a penetration angle of $75{}^{\circ }$ results in much less drug reaching the macula, see Figure 7. During the 30 days, we calculated about 31% more drug on the macula at a penetration angle of $50{}^{\circ }$. In [33], the differences in the amount of drug reaching the macula between an injection angle of $75{}^{\circ }$ and of $50{}^{\circ }$ are even around 50%, because there an injection with a fast injection speed was considered. We neglect the injection velocity in our model, since this would lead to a different type of equations and thus to different solution methods. With our model, we only analyze the effect of drug diffusion and not drug convection through the injection.

In all cases considered, the concentration of the drug at the macula increases in the first days. The highest concentration is reached after about 6 days, and after that we measure a monotonic decrease of the concentration. The maximum is reached at $0.282\frac{\mathrm{kg}}{m^{3}}$ for a penetration angle of $50{}^{\circ }$ and $0.214\frac{\mathrm{kg}}{m^{3}}$ for $75{}^{\circ }$. The penetration angle of $25{}^{\circ }$ leads to a similar result as that for $50{}^{\circ }$. We obtained about 2% more drug on the macula for a penetration angle of $25{}^{\circ }$ and the highest concentration is $0.291\frac{\mathrm{kg}}{m^{3}}$. At both $25{}^{\circ }$ and $50{}^{\circ }$ angles, the injection needle is almost exactly aligned with the macula, which explains the positive results. In contrast, at an angle of $75{}^{\circ }$, the drug is injected farther from the macula and closer to the lens. This results in more drug being cleared through the retina before it can reach the macula. This effect is confirmed by analyzing the functionals $J_{R}$ and $J_{M,\Omega }$, see Figure 8.

We calculate that at a penetration angle of $75{}^{\circ }$, about 30% of the injected drug reaches the macula and that about 70% diffuses through the retina and is washed out by the blood vessels. The other two angles $25{}^{\circ }$ and $50{}^{\circ }$ show that 40% of the drug reaches the macula and 60% of the drug diffuses through the retina. Furthermore, the functional $J_{\Omega }$ shows us that more drug remains in the vitreous body for these two penetration angles for all times than at $75{}^{\circ }$. For $25{}^{\circ }$, we measure slightly higher concentration of drug in the vitreous than for $50{}^{\circ }$.

In summary, an injection angle $\psi_{xy}$ between $25{}^{\circ }$ and $50{}^{\circ }$ gives similar good results and demonstrates that its influence on drug distribution is small. With an injection angle in this range, a greater amount of drug reaches the macula than with larger injection angles. However, an injection with $\psi_{xy}=75{}^{\circ }$ leads to changes in the amount of drug around the macula that are likely to have a negative effect on the therapy; therapy will be less effective in this case.

When examining the penetration angle $\psi_{z}$, the simulations show that the larger the angle, the less drug is around the macula. This result can be explained by the fact that a larger amount of the drug escapes through the retina and cannot reach the site of action. In Figure 9, the penetration angle $\psi_{xy}$ is chosen to be $50{}^{\circ }$. At a penetration angle $\psi_{z}=115{}^{\circ }$, we measure 10% less drug on the macula than at $\psi_{z}=90{}^{\circ }$ and as much as 38% less at $\psi_{z}=140{}^{\circ }$. In addition, a penetration angle of $\psi_{z}=90{}^{\circ }$ results in a loss of about 60% of the injected drug through the retina, an angle of $\psi_{z}=115{}^{\circ }$ of about 65%, and an angle of $\psi_{z}=140{}^{\circ }$ of about 75%. This suggests that changes in the penetration angle $\psi_{z}$ may limit the local effect of the drug as a VEGF blocker at the macula.

### 3.4. The Influence of the Diffusion Coefficient on Drug Distribution

In age-related macular degeneration, an anti-VEGF drug is injected as standard. The choice of drug determines the course of therapy. Currently, the most commonly used anti-VEGF drugs are ranibizumab (Lucentis, Genentech, San Francisco, CA, USA and Novartis Ophthalmics, Basel, Switzerland), bevacizumab (Avastin, Genentech, San Francisco, CA, USA) and aflibercept (Eylea, Regeneron, Tarrytown, NY, USA). There are a large number of studies and comparisons of these drugs for the treatment of age-related macular degeneration, diabetic retinopathy, diabetic macular edema and retinal vein occlusions. For example, in [49,50,51] all three anti-VEGF drugs are compared in patients with diabetic macular edema. All three drugs result in improvement in visual acuity. However, it is unclear which drug is best suited for which patient. In age-related macular degeneration, roughly equivalent effects on the healing process are observed for the drugs bevacizumab and ranibizumab in [52]. Aflibercept is a newer drug used to treat age-related macular degeneration. In [53], it is reported that longer intervals between injections can be achieved with aflibercept. The most important process for the effectiveness of therapy is pharmacology. We consider this in our models and simulations in another project. In this paper, we analyze the effect of drug diffusion in the three-dimensional vitreous, i.e., how much drug reaches the site of action. Our virtual experiments are designed with fixed parameters for eye geometry and for injection. With the mathematical functionals we have developed, we can evaluate the amount of drug used in the vitreous and around the macula over time, so we can perform a systematic analysis to study the influence of certain parameters on drug distribution. One very relevant parameter is the diffusion coefficient, as already recognized in [30]. The molecular weight of bevacizumab is 148 kDa, so it is a large molecule with a half-life twice as long as ranibizumab [50,51]. The molecular weight of ranibizumab is 48 kDa [54] and of aflibercept 115 kDA, which also has higher affinity than ranibizumab or bevacizumab [55,56,57].

In the following, we perform simulations for the diffusion parameter $D=4\times 10^{-11}\frac{m^{2}}{s}$ from [35] and for some fictive diffusion parameters, one smaller, $D=2\times 10^{-11}\frac{m^{2}}{s}$, and one larger, $D=8\times 10^{-11}\frac{m^{2}}{s}$, to quantify the differences. This gives us insight into drug diffusion and conclusions can be drawn for other diffusion coefficients as well. First, we consider the comparison of drug concentration in specific regions of the eye as illustrated in Figure 10.

The larger the diffusion coefficient, the faster the drug diffuses and reaches the macula more quickly. Heavier molecules have smaller diffusion coefficients. However, after a few days, only a small amount of the drug is observed around the macula. At the macula, a concentration greater than $0.1\frac{\mathrm{kg}}{m^{3}}$ can be achieved in the first days only with the larger diffusion coefficient, $D=8\times 10^{-11}\frac{m^{2}}{s}$. The time period is about 10 days, then the concentration at the macula decreases. For molecules with a slower diffusion coefficient, $D=2\times 10^{-11}\frac{m^{2}}{s}$, this period is about 22 days and about 15 days for $D=4\times 10^{-11}\frac{m^{2}}{s}$. In contrast, the highest concentration is obtained for the largest diffusion coefficient. The maximum concentration $2.9\frac{\mathrm{kg}}{m^{3}}$ is reached for $D=8\times 10^{-11}\frac{m^{2}}{s}$, $2.8\frac{\mathrm{kg}}{m^{3}}$ for $D=4\times 10^{-11}\frac{m^{2}}{s}$ and $2.7\frac{\mathrm{kg}}{m^{3}}$ for $D=2\times 10^{-11}\frac{m^{2}}{s}$. Furthermore, a greater amount of drug reaches the macula over the 30 days when the diffusion coefficient is smaller. After 30 days, for $D=2\times 10^{-11}\frac{m^{2}}{s}$ about 50% of the drug escapes through the retina, about 40% can act on the macula and 10% is still in the vitreous body. For $D=8\times 10^{-11}\frac{m^{2}}{s}$ about 80% of the drug escapes through the retina and only 20% acts on the macula, leaving nothing in the vitreous. The results for $D=4\times 10^{-11}\frac{m^{2}}{s}$ are in between the values obtained for the previously mentioned cases. Overall, a higher diffusion coefficient results in a significant loss of drug through the retina and less drug can reach the macula.

## 4. Discussion

In this paper, we discuss the efficacy of anti-VEGF therapy against wet age-related macular degeneration. For this purpose, we model the drug distribution in the vitreous with a convection-diffusion equation. The convection is caused by the vitreous humor flow, which is modeled with the Darcy equation, taking into account the consistency of the vitreous, including the collagen network present within. Mathematically challenging are the mixed boundary conditions describing realistic physiology in the complex geometry of the eye, so even the Darcy equation here takes on a new face not found in the literature, an inflow is prescribed by a Poiseuille velocity profile. The derived model is solved numerically using the Finite Element method. For this purpose, over 6000 lines of code were implemented in the existing deal.ii environment. Trilinear Lagrangian elements are used to discretize the drug concentration, Raviart-Thomas elements are used for the velocity of the vitreous humor, and discontinuous piecewise constant elements are used for the associated pressure. The system is solved in a decoupled way, first solving the Darcy equation using the Schur complement method and the CG method, and then solving the time-dependent convection-diffusion equation using the Rothe method and the GMRES method with an ILU preconditioner. To quantitatively evaluate the efficacy of therapy, we introduce functionals, the relative amount of drug remaining in the vitreous at a given time point, the amount of drug localized around the macula, and the total amount of drug available to react with VEGF molecules over a given time. We perform long-term simulations covering the period of one injection, i.e., 30 days.

We discuss the influence of gravity on drug distribution and found that unlike the experiments performed in [37], where the gravity causes a higher drug concentration at the bottom of the test tube than at the top, even after 7 days, that gravity does not play a significant role in our models and simulations. Furthermore, we compared 4 injection positions and found that the standard position used in the treatment performed worst, namely then the least amount of drug arrived at the macula. An injection into the center of the vitreous, 10 mm from the limbus via pars plana toward the center of the posterior pole with needle depth about 10 mm, seems to be optimal, so that the highest drug amount reaches the macula in 30 days. Our results are purely theoretical; if the injection positions considered are reasonable from a medical point of view is another task that cannot be discussed here. The optimal injection angles are $25{}^{\circ }$ between the needle direction and the optical axis in the xy-plane, and $90{}^{\circ }$ between the needle direction and the z-axis. The differences between $25{}^{\circ }$ and $50{}^{\circ }$ are small, while the differences to $75{}^{\circ }$ are large. The highest drug concentration at the macula occurs 6 days after injection. The simulations are also useful to try different diffusion constants and quantify the difference in the functionals. The following values were compared: $2\times 10^{-11}\;$
$m^{2}$/s, 4$\times 10^{-11}\;$
$m^{2}$/s and $8\times 10^{-11}\;$
$m^{2}$/s, where $4\times 10^{-11}\;$
$m^{2}$/s corresponds to bevacizumab. A smaller diffusion coefficient, corresponding to heavier molecules, appears to be beneficial for treatment. The drug would not diffuse through the vitreous as quickly, allowing more of it to be transported to the macula by the vitreous humor flow.

Our model is certainly already very complex and realistic, but there are further possibilities for extension. In a current project, we are analyzing different physical boundary conditions for the drug at the hyaloid membrane to also see the distribution in the anterior chamber. We know that a large fraction of the drug is washed out through Schlemm’s canal. In [41], we consider a viscoelastic vitreous model and thus include the collagen fibers and the consistency of the vitreous. A fluid structure interaction model is also created to include the elastic sclera and lens. A recently completed dissertation [58] covers the current pharmacology of anti-VEGF therapy, including the life span of the drug. It would also be useful to calibrate and validate the models with experimental data. This will require close collaboration with experimentalists. It should also be noted that simulations of the different models require different numerical methods that take time to be implemented. However, once developed, numerous tests are readily available to provide insights into the physical and chemical processes that can improve therapy. The in silico experiments proposed here can be used together with studies, in vitro and animal experiments. However, they can drastically reduce the number and thus the cost of these experiments, since numerous test can be performed and preselected by the computer and its user.

## 5. Conclusions

It has been demonstrated that VEGF is only produced locally and also only acts locally [59]. It is therefore important that the drug reaches the site of action, the macula. It is also important that enough molecules reach this area so that the VEGF is completely blocked. Since VEGF is also produced continuously, it is important that a sufficient concentration of drug reaches the macula over a long period of time. The drug distribution and amount of drug on the macula can be influenced by the choice of drug and the way of administration. Based on our models and simulation results, we can recommend a refined AMD therapy. Whether the findings are useful from a medical point of view and feasible in practice remains an open question at present and can be communicated at a later time.


**Refined Therapy:**
When the drug is injected centrally into the vitreous, a certain amount of the drug reaches the macula the longest. This can be interesting for longer acting drugs, e.g., aflibercept. Otherwise, a large portion of the drug escapes through the retina.If the drug is injected closer to the macula, a higher concentration arrives there, but for a shorter period of time. This can be interesting for an intensified initial treatment, e.g., for the treatment of a thick edema.The needle should be oriented in the direction of the macula. An unfavorable insertion angle can lead to a loss of up to 38% of the drug at the macula.A larger diffusion coefficient for the drug, a lighter molecule, results in a higher drug concentration at the macula, but on average over 30 days, it results in a lower drug concentration at the macula because more drug also escapes through the retina more quickly.


## 6. Patents

E. Friedmann, S. Dörsam, and V. Olkhovskiy, Test and optimization of medical treatments for the human eye. European patent application ep 18000349, 2018.

## Figures and Tables

**Figure 1 diagnostics-13-00975-f001:**
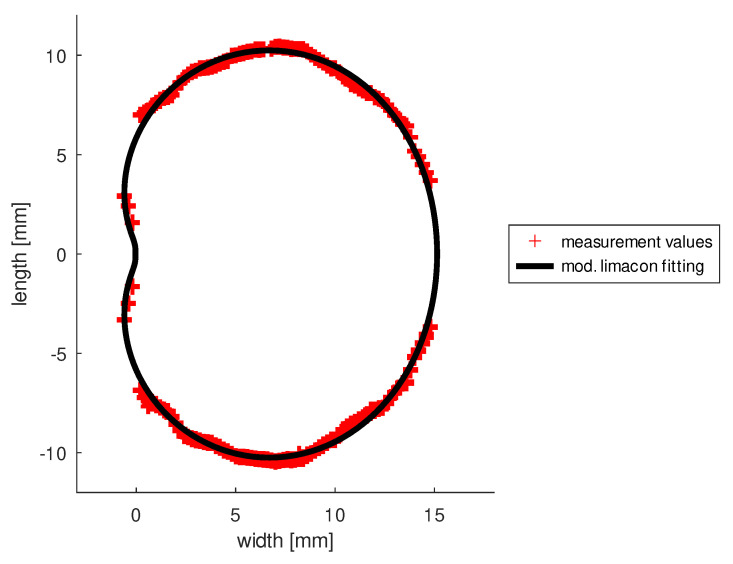
The shape of the human vitreous described with a mathematical function, the modified Limacon, obtained by fitting the US measurement data from 12 patients.

**Figure 2 diagnostics-13-00975-f002:**
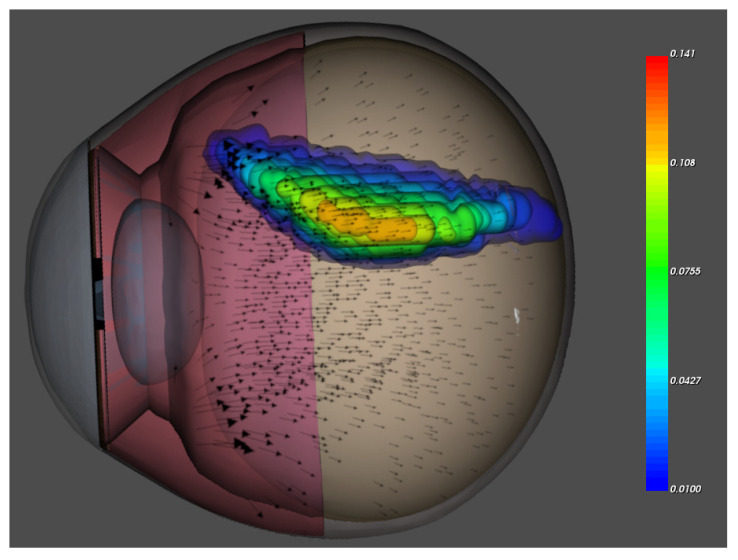
Our Virtual Eye with cornea, sclera, iris, lens, ciliary body, vitreous and retina; the drug concentration at a given time visualized with isosurfaces and the vitreous humor flow represented with arrows.

**Figure 3 diagnostics-13-00975-f003:**
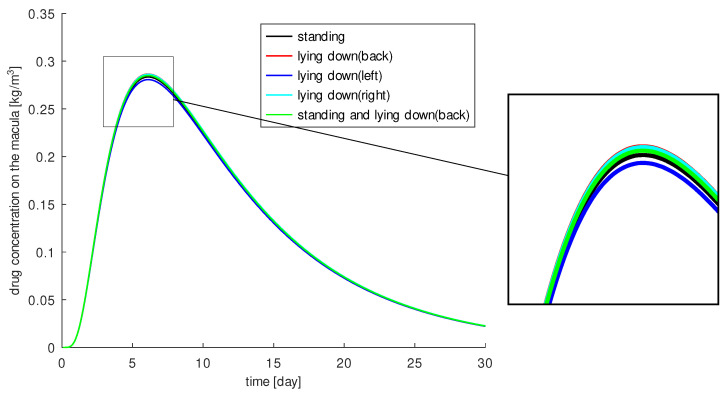
The drug amount around the macula for different head positions: standing (black), lying down on the back (red), lying down on the left side (blue), lying down on the right side (light blue), and standing half the day and lying down on the back the other half of the day (green).

**Figure 4 diagnostics-13-00975-f004:**
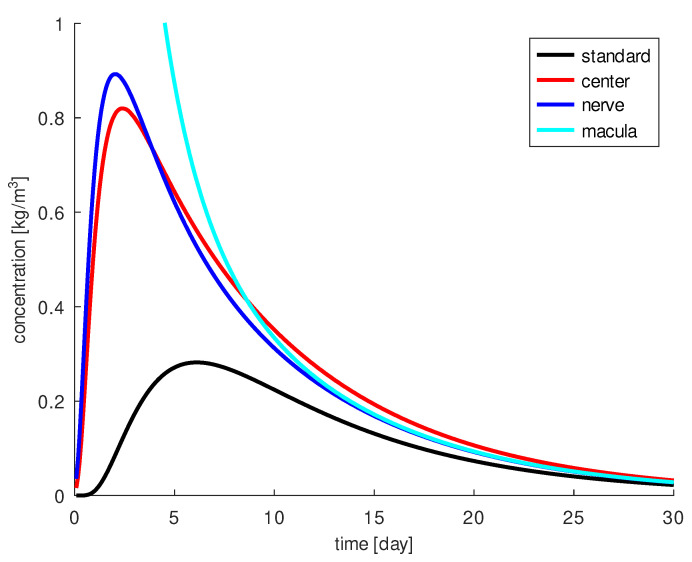
We test four different injection positions. Position 1 (called standard) is the standard position for an injection in the left eye: 3.5 mm from the limbus via the pars plana direction to the center of the posterior pole, needle depth about 5 mm; Position 2 (called center) is 10 mm from the limbus via pars plana direction to the center of the posterior pole, needle depth about 10 mm; Position 3 (called nerve) is 8 mm from the limbus via pars plana direction laterally to the posterior pole and the needle penetrates 7–8 mm; Position 4 (called macula) is chosen 10 mm from the limbus via the pars plana direction to the center of the posterior pole with an 18 mm needle (injection site near the macula) penetrating in the direction of the optic nerve.

**Figure 5 diagnostics-13-00975-f005:**
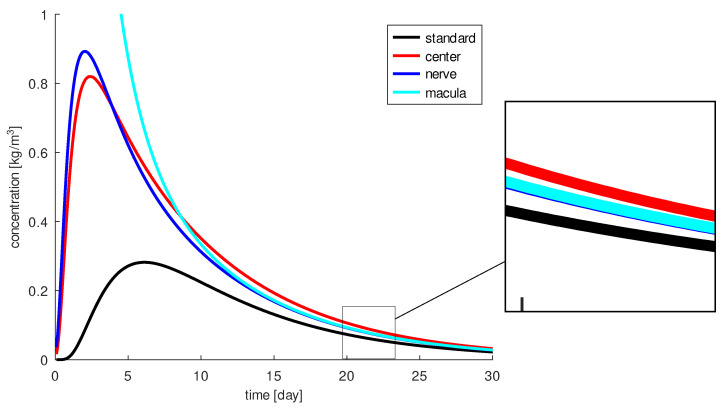
The drug concentration at the macula for different injection positions over time: the trajectory for the standard injection (black), for the injection into the center of the vitreous (red), for the injection near the optic nerve (blue) and for the injection near the macula (light blue).

**Figure 6 diagnostics-13-00975-f006:**
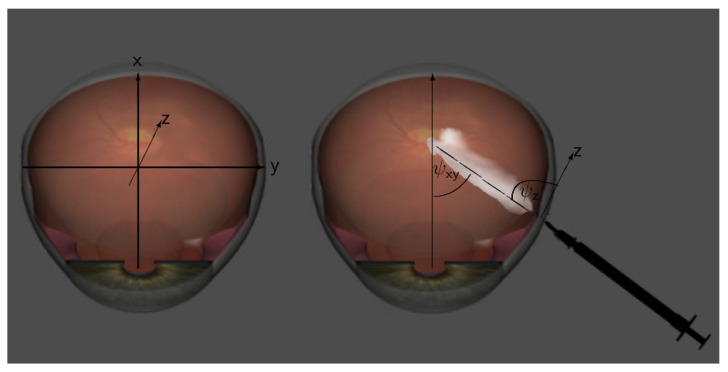
On the left, our coordinate system in our Virtual Eye, and on the right, a schematic representation of the two penetration angles $\psi_{xy}$ and $\psi_{z}$ of the injection for orientation in space.

**Figure 7 diagnostics-13-00975-f007:**
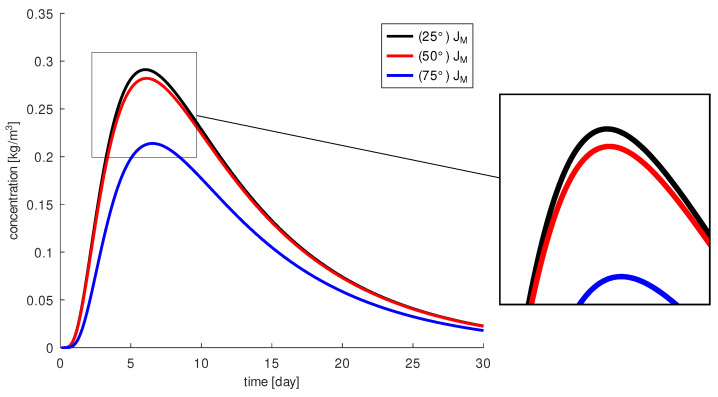
The amount of drug concentration at the macula, $J_{M}$, depending on the penetration angle $\psi_{xy}$: $25{}^{\circ }$ (black), $50{}^{\circ }$ (red) and $75{}^{\circ }$ (blue).

**Figure 8 diagnostics-13-00975-f008:**
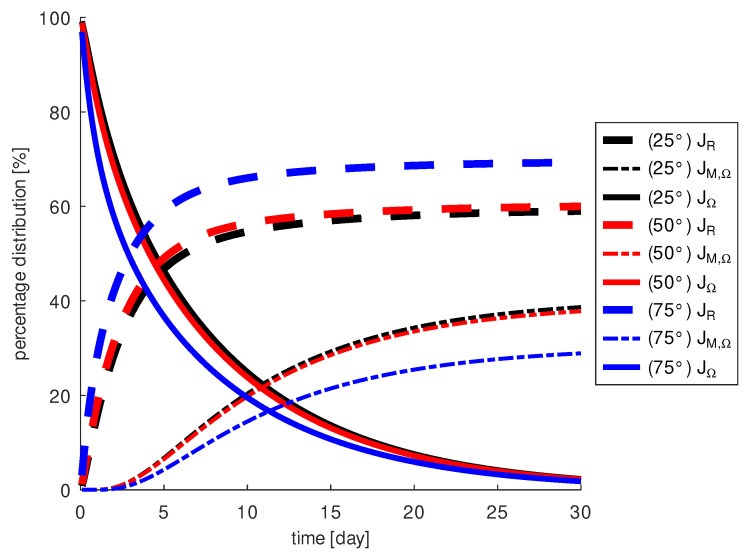
The fraction of the injected drug in the vitreous, $J_{\Omega }$, at the macula, $J_{M,\Omega }$, and at the retina, $J_{R}$, depending on the penetration angle $\psi_{xy}$.

**Figure 9 diagnostics-13-00975-f009:**
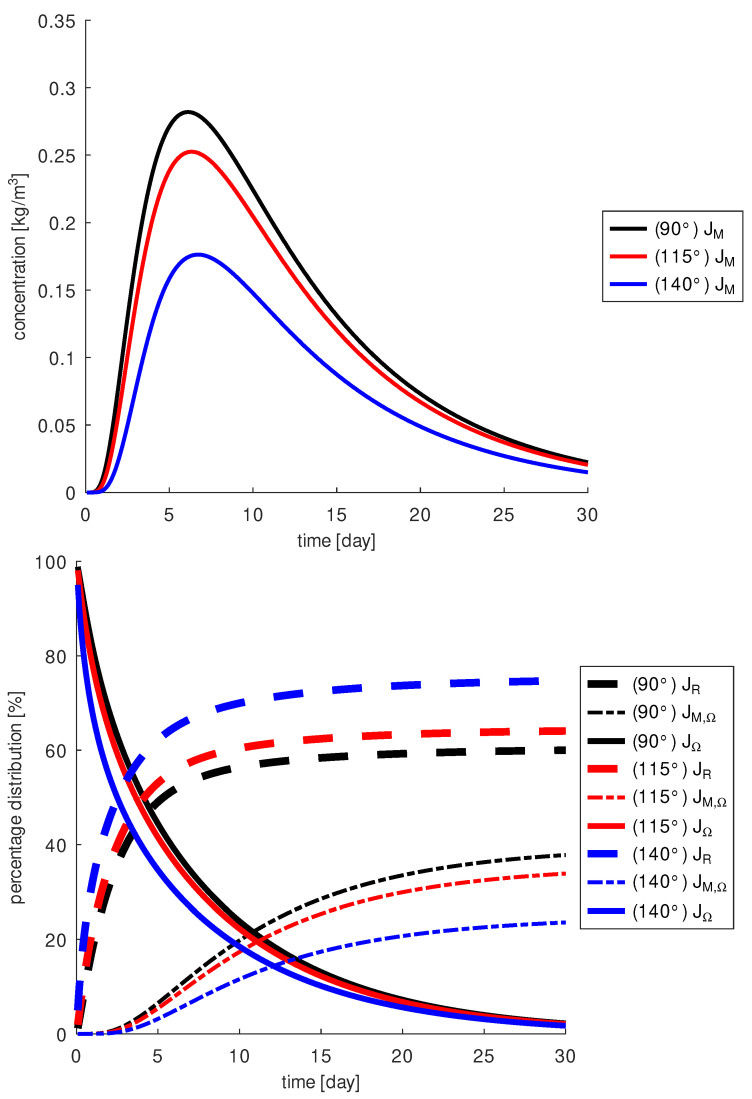
Top: drug concentration at the macula, $J_{M}$ and bottom: fraction of injected drug in the vitreous, $J_{\Omega }$, around the macula, $J_{M,\Omega }$, and lost through the retina, $J_{R}$, as a function of penetration angle $\psi_{z}$: with an injection angle of $90{}^{\circ }$ (black) the largest amount of drug is inside the vitreous (black solid line) and also at the macula (black dashed dotted line).

**Figure 10 diagnostics-13-00975-f010:**
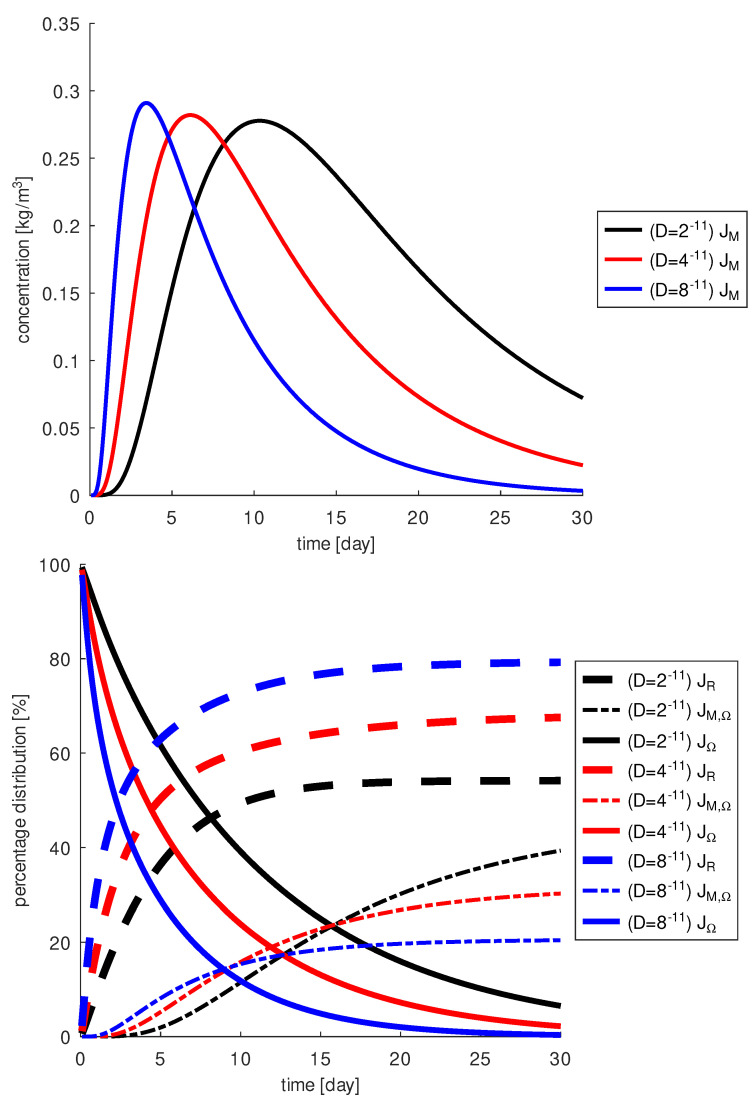
Top: the drug amount at the macula, $J_{M}$, as a function of *D*, lighter molecules (blue) are faster; bottom: the drug amount in the vitreous, $J_{\Omega }$, at the macula, $J_{M,\Omega }$, and lost through the retina, $J_{R}$, in dependence of *D*. Heavier molecules remain longer in the vitreous, the loss through the retina is less than for lighter and faster molecules, and more of them finally reach the macula after 17 days, in the first 8 days more lighter molecules reach the macula and between 8 and 16 days more intermediate weight molecules reach the macula.

## Data Availability

Data are contained within the article. Further virtual experiments can be performed by the authors after request.

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
