# Peer review of "Models and Algorithms for the Refinement of Therapeutic Approaches for Retinal Diseases"

_diagnostics, 2023, doi:10.3390/diagnostics13050975_

Round 1

Reviewer 1 Report

1.     Highlight the article innovation point in the abstract. Why should anyone read your article?

2.     The Abstract should contain answers to the following questions: What problem was studied and why is it important? What methods were used? What are the important results? What conclusions can be drawn from the results? What is the novelty of the work and where does it go beyond previous efforts in the literature? Please include specific and quantitative results in your Abstract, while ensuring that it is suitable for a broad audience. 

3.     Brief description of each condition and assumption employed need to be addressed clearly.

4.     It is helpful to complete the description of how to collect data, data processing scenarios, and interpret the data collection.

5.     Explain used Numerical scheme for solutions in a new section and few advantages of it.

6.     Revise the introduction such that each paragraph shall present the meaning of a concept/keyword. Update with recent publications

7.     Figures: First, the self-explanatory legend is required. There is not enough info on the drawings. That means, I would like to see more connection between the drawings and the model including boundary conditions. Finally, by looking into the figure it would be good to get some visual representation of the physical process the authors consider.

8.     How to reliable on your model.

9.     Use vector graphic images, and avoid serif fonts in figures (use sans-serif types).

Reviewer 2 Report

This is a very interesting article concerning possible new models and algorithms for the refinement of the intravitreal therapy in retinal diseases.

This is a well-written article and it is very suitable for this special issue.

I only suggest two things:

1) to check the English throughout the manuscript, because there are some spelling mistakes.

2) to mention the latest drugs Brolucizumab and Faricimab in the Introduction section, even if they were not analyzed in this study.

Author Response

Reviewer 2: This is a very interesting article concerning possible new models and algorithms for the refinement of the intravitreal therapy in retinal diseases.

This is a well-written article and it is very suitable for this special issue.

I only suggest two things:

1) to check the English throughout the manuscript, because there are some spelling mistakes.

I completely revised the language of the entire article.

2) to mention the latest drugs Brolucizumab and Faricimab in the Introduction section, even if they were not analyzed in this study.

Thank you for this very helpful comment. The new drugs were included in the introduction, and we mentioned also the promissing treatment with a Port Delivery System filled with a drug that can be refilled or with drug loaded hydrogels for sustained delivery. Of course, this information has a big impact on the topicality of the article. We added also some recent literature. Follwing change was made in the introduction:

Since then, other more effective anti-VEGF agents have been developed: Lucentis (ranibizumab), Avastin (bevacizumab), Eylea (aflibercept), and the latest approved agents Beovu (brolucizumab, 2020) [22] and Vabysmo (faricimab, 2022) [23]. Treatment is not fixed; there are four dosing regimens in clinical use [21]. Monthly or bimonthly injections lead to long-term success, but do not take into account individual disease progression. A pro-re-nata strategy requires many check-ups and does not show good improvement, as reactivation of the disease is often detected too late, which can lead to irreversible visual loss. A treat-and-extend regimen is a personalized treatment in which the next injection is optimally planned [24,25]. An observe-and-plan regimen is also a prosonalized treatment that includes a personalized treatment plan with multiple injections [26]. Of course, the newly developed drugs are more effective and therefore require fewer injections. There are still many unresolved issues regarding frequency of treatment, proper dose, location of injection, effectiveness of therapy, choice of drug and more. Nevertheless, due to the high cost of the drugs and the inconvenience of the injections, as well as the increasing burden on patients due to the many consultations and inconvenient injections, there is an urgent need for a peronalized, long-lasting optimized therapy solution for patients with AMD. One promissing aproach is treatment with a Port Delivery System filled with a drug that can be refilled [27] or with drug loaded hydrogels for sustained delivery [28,29].

Thank you very much for your time and helpful comments!

Round 2

Reviewer 1 Report

The authors addressed the referees points effectively.